# Evaluation of the Use of a Coffee Industry By-Product in a Cereal-Based Extruded Food Product

**DOI:** 10.3390/foods9081008

**Published:** 2020-07-27

**Authors:** Elisa A. Beltrán-Medina, Guadalupe M. Guatemala-Morales, Eduardo Padilla-Camberos, Rosa I. Corona-González, Pedro M. Mondragón-Cortez, Enrique Arriola-Guevara

**Affiliations:** 1Tecnología Alimentaria, Biotecnología Médica y Farmacéutica, Centro de Investigación y Asistencia en Tecnología y Diseño del Estado de Jalisco, A.C. (CIATEJ), Normalistas 800, C.P. 44270 Guadalajara, Jalisco, Mexico; es_ebeltran@ciatej.mx (E.A.B.-M.); gguatemala@ciatej.mx (G.M.G.-M.); pmondragon@ciatej.mx (P.M.M.-C.); 2Departamento de Ingeniería Química, Centro Universitario de Ciencias Exactas e Ingenierías, Universidad de Guadalajara. Blvd. Marcelino García Barragán #1421, esq. Calzada Olímpica. C.P. 44430 Guadalajara, Jalisco, Mexico; rosa.corona@academicos.udg.mx

**Keywords:** coffee silverskin, chemical characterization, toxicological analysis, extrusion, extreme vertices mixture design, product development

## Abstract

The evaluation of by-products to be added to food products is complex, as the residues must be analyzed to demonstrate their potential use as safe foods, as well as to propose the appropriate process and product for recycling. Since coffee is a very popular beverage worldwide, the coffee industry is responsible for generating large amounts of by-products, which include the coffee silverskin (CS), the only by-product of the roasting process. In this work, its characterization and food safety were evaluated by chemical composition assays, microbiological determinations, aflatoxin measurements and acute toxicity tests. The results showed that CS is safe for use in food, in addition to providing dietary fiber, protein and bioactive compounds. An extruded cereal-based ready-to-eat food product was developed through an extreme vertices mixture design, producing an extruded food product being a source of protein and with a high fiber content. Up to 15% of CS was incorporated in the extruded product. This work contributes to the establishment of routes for the valorization of CS; nevertheless, further research is necessary to demonstrate the sustainability of this food industry by-product.

## 1. Introduction

Today, there is a considerable emphasis on the recovery, recycling and upgrading of wastes, particularly in the food processing industry, in which wastes, effluents, residues and by-products can be recovered. They can often be upgraded into useful products and value-added food supplements that can provide dietary fiber and bioactive compounds [1,2]. The possibility of the utilization of these food processing by-products for manufacturing various human foods has created enormous scope for waste reduction, indirect income generation, the reduction of raw material costs [1,3] and even the potential for them to be considered as novel foods with beneficial properties [4]. However, for the development of future sustainable industrial processes centered on the valorization of food waste, aspects such as technical feasibility, an analysis of their-economic potential and a life-cycle-based environmental assessment need to be considered [5].

Coffee is one of the most consumed beverages in the world and is the second largest traded commodity after petroleum [6]. The coffee production chain begins with the harvest of the ripe coffee berries that are to be treated in order to separate the pulp from the coffee bean by one of two processes—(a) a wet process or (b) a dry process—where the green coffee bean is obtained. Finally, the bean is heat treated by a process called roasting, thus producing the coffee that will be used for the preparation of the drink [4,7]. In the world, during the 2018/2019 season, 10.3 million tons of green coffee was produced [8]. Since coffee is a very popular and appreciated beverage around the world, the coffee industry is responsible for generating large amounts of wastes, which include the coffee silverskin (CS), the only waste obtained during the roasting process [7,9]. The CS represents about 4.2% (w/w) of the coffee beans [9]. Despite the produced quantity being low compared to that of other coffee by-products, it has been reported that for 120 tons of roasted coffee, about 1 ton of CS is produced [10]. It can be considered that if all the green coffee produced worldwide during the 2018/19 season had been roasted, it would be equivalent to having produced around 71,822 tons of CS (conversion factor: 1.19 tons of green coffee = 1 ton of roasted coffee [8]). CS is a yellowish transparent endosperm that covers each green coffee bean (Figure 1) [4,7,9] and is currently used as a fuel and fertilizer [11]. However, coffee wastes have been reported to possess bioactive compounds, mainly secondary metabolites such as phenolic acids, for example, hydroxycinnamic acids and flavonoids, desired for their beneficial antioxidant properties [12,13]. 5-caffeoylquinic acid (5CQA) belongs to the family of the chlorogenic acids (hydroxycinnamic acids). It is one of the most abundant polyphenolic compounds in the human diet and is produced by certain plant species; it is an important component in coffee and in the CS [14,15]. 5CQA is of special interest due to the wide spectrum of its potentially beneficial effects on health, including antidiabetic, anti-obesity, antioxidant, anti-hypertension, anti-inflammatory and antibacterial effects [16,17]. 

CS has been reported as a source of chlorogenic acids; however, to date, there are few reports concerning the content of 5CQA in CS, and those that exist show controversial results, since the reported concentrations are in the range of 1000 to 11,678 mg of 5CQA/kg of CS [11,18,19,20]. Different studies have shown the functional properties of CS such as a high dietary fiber content (54.11 to 74.15 g/100 g of CS) [9,21,22] and a total phenolic content in the range of 4.6 to 46.65 mg/g, depending on the extraction method employed [11,21,23,24]. The principal constituents of its fibrous tissues are cellulose (24%) and hemicellulose (17%). It is a source of minerals such as potassium (21,100 mg/kg dry basis (db)), iron (843 mg/kg db), sodium (57 mg/kg db), manganese (50 mg/kg db) and zinc (22 mg/kg db), among others [9]. The enzyme inhibitory properties of CS extracts and peptide composition of CS protein hydrolysates have been investigated, from the perspective of their application in the pharmaceutical and nutraceutical industry [24,25].

The holistic concept of food production tries to connect differing goals, such as the highest product quality and safety, highest production efficiency and the integration of environmental aspects into product development and food production. Vegetable residues mostly contain considerable amounts of potentially interesting compounds [1]. However, the benefits of recycling should not be undermined by the environmental impacts caused by new production processes [26]. Food extrusion is a versatile process in food engineering as it combines various unit operations such as transport, thermomechanical and degradation changes, mixing and molding. It is a technology widely used in the food industry due to its versatility, high productivity and energy efficiency [27,28]. Extrusion cooking is increasingly used in the food industries for the development of new cereal-based snacks, baby foods and breakfast cereals [29]. The incorporation of by-products from different fruit and vegetable processing industries into extruded products has led to hope for their utilization as well as the development of nutritionally healthy extruded products [30]. The extrusion process has been used to develop new products in which 2% to 20% of various by-products of the agri-food industry have been incorporated, such as barley-fruit and cauliflower by-products and red lentil-carrot pomace, among others [30,31,32,33]. The by-product incorporation in extruded food has been reported at the lab scale, and no industrial-level studies have been shown. Nevertheless, extrusion is a mature and scalable technology; even scale-up considerations and mathematical models for extrusion cooking are available [34,35].

Cereal-based food products have been the basis of the human diet since ancient times. Cereals contain all the macronutrients (protein, fats and carbohydrates) we need for support and maintenance [36]. They contain only low levels of micronutrients, most of which are lost during processing for food [37], bringing the possibility to incorporate new raw materials that provide these micronutrients. To the best of our knowledge, there are few reports on the use of CS in a cereal-based product. A treatment of CS with alkaline hydrogen peroxide before being added to Barbari bread to improve its properties has been described [38]. In another study, CS was added to cake as a fat replacer [39]. Some authors investigated the use of CS in biscuits, where it was incorporated as a sugar replacer or to enhance the phenolic content and antioxidant capacity of the product, employing a standardized formulation [40,41]. However, no studies regarding the formulation design under official standards to create a product with specific requirements have been reported. Hence, the development of a cereal-based food product adding CS using extrusion technology is proposed.

In all those articles in which CS was added in cereal-based products, wheat flour was used as the cereal basis [38,39,40,41]; nonetheless, in present work, corn and popped amaranth were chosen as the cereal product base. Corn is undoubtedly part of the identity of Mexico; it is present in the daily life of its inhabitants [42], which will allow the obtaining of a familiar flavor and better acceptance of the developed product. Amaranth is a popular snack in Mexico of pre-Hispanic origin [42] and is characterized by an excellent nutritional composition (*A. hypochondriacus,* protein, 15.9%; lysine, 4.9 g/100 g of protein; fat, 6.1%; tocopherols, 5.5 mg/100 g; starch, 62.4%; sucrose, 1.4%; ash, 3.3% [43]); nevertheless, it is difficult to produce expanded products directly by the extrusion cooking of amaranth grain alone because of its high fat content. Therefore, the extrusion cooking of amaranth flour in combination with other cereals produces well-accepted forms of expanded extrudates [43]. The combination of these cereals with the CS could allow the obtaining of a product with good nutritional quality.

Therefore, as CS appears to be a potential new low-cost ingredient, the aim of this work was to evaluate whether its consumption was harmless to humans, demonstrating the food safety of CS by microbiological tests and the determination of aflatoxins and the Lethal Dose (LD50) by the acute oral toxicity test. Its potential use as a food ingredient was evaluated by determining its nutritional contribution and by developing a cereal-based extruded food product with this new ingredient added.

## 2. Materials and Methods

### 2.1. Materials

CS produced by roasting coffee beans (*C. Arabica* 100%) was obtained from two states of Mexico (Chiapas and Jalisco). Popped amaranth was purchased from Nutriactivate Company (Puebla, Puebla, Mexico), and white corn was obtained from the food market of Guadalajara city (Jalisco, Mexico). CS, popped amaranth and white corn were milled prior to the extraction and extrusion process (Average Particle Size = 0.28 ± 0.01 mm). 

5-caffeoylquinic acid powder reference standard (USP 12601), 2,2-diphenyl-1-picrylhydrazil (DPPH), 2,2′-azino-bis(3-ethylbenzthiazoline-6-sulphonic acid) (ABTS) and 6-hydroxy-2,5,7,8-tetramethyl chroman-2-carboxylic acid (Trolox) were purchased from Sigma-Aldrich (St. Louis, MO, USA). Methanol (MeOH) was obtained at HPLC grade (Sigma-Aldrich, St. Louis, MO, USA). Phosphoric acid was obtained at reagent grade (Karal, Leon, Guanajuato, Mexico).

The standard of 5CQA was diluted in MeOH to obtain a stock solution at 1000 µg/mL, from which the calibration curve was prepared. All the solutions remained refrigerated at 4 °C in amber vials.

### 2.2. CS Characterization and Microbiological Quality

#### 2.2.1. Bromatological and Microbiological Analysis

The bromatological analyses were performed according to the following Mexican Regulations: moisture, NMX-F-083-1986 [44]; protein, NMX-F-608-NORMEX-2011 [45]; ashes, NMX-F-607-NORMEX-2013 [46]; fats (ethereal extract), NOM-086-SSA1-1994 (Regulatory Appendix C, Number 1) [47]; and, carbohydrates, method 986.25 A.O.A.C. Volume 1 [48].

The microbiological analysis was performed according the Official Mexican Regulations: aerobic mesophilic bacteria, NOM-092-SSA1-1994 [49]; total coliforms, NOM-113-SSA1-1994 [50]; molds and yeasts, NOM-111-SSA1-1994 [51]; *Salmonella*, NOM-114-SSA1-1994 [52]; *Escherichia coli*, CCAYAC-M-004 [53]; and *Staphylococcus aureus*, NOM-115-SSA1-1994 [54].

#### 2.2.2. Total Dietary Fiber (TDF)

The TDF was estimated by the enzymatic gravimetric method according to the Mexican Regulation NMX-F-622-NORMEX-2008 [55]. Briefly, one gram of sample suspended in phosphate buffer solution was sequentially digested by heat stable α-amylase for 30 min in a boiling water bath, after which a 0.275 M NaOH solution and protease were added and incubated for 30 min at 60 °C. Then, a 0.325 M HCl solution and amyloglucosidase were added and incubated for 30 min at 60 °C. After filtration, the insoluble dietary fiber was recovered from enzyme digestate, dried at room temperature and then weighed. Soluble dietary fiber in the filtrate was precipitated with ethanol and filtered. The precipitate was dried and weighed. Insoluble and soluble dietary fiber contents were corrected for residual protein and ash content. The TDF content was the sum of both fibers.

#### 2.2.3. Extraction Method

This extraction method was used for DPPH, ABTS, total polyphenol and HPLC analysis. The extraction method was adapted from Del Río et al. (2014) [56]; 0.5 g of sample (CS or extruded product) was weighed, and 5 mL of MeOH/water 3:1 (v/v) was added. The mixture was sonicated (Branson 5800, Dansbury, CT, USA) at a 40 kHz frequency for 15 min, removed and stirred in a Vortex-Genie (Scientific Industries, Bohemia, NY, USA) for another 15 min; afterwards, it was centrifuged at 3400 rpm for 10 min. The supernatant was transferred to another container, and the residue was re-extracted. The second extract was added to the first, and it was filtered (0.45 µm). All the extracts were kept in amber vials, under refrigeration, until analysis.

#### 2.2.4. Total Polyphenol Determination

The quantification of total polyphenols was carried out by the Folin–Ciocalteu method proposed by Singleton and Rossi (1965) [57]. The extracts (30 µL) were mixed with 150 µL of Folin–Ciocalteu reagent (1:10), followed by the addition of 120 µL of 20% (w/v) sodium carbonate. After 1 h, the absorbance at 760 nm was read in the spectrophotometer. The results are expressed as g gallic acid equivalents (GAE)/100 g sample.

#### 2.2.5. Antioxidant Activity

The antioxidant activity of the extracts was determined by two methods: the ABTS and DPPH (free radical scavenging) assays. The ABTS assay was based on a method developed by Nenadis et al. (2004) [58]. A solution of 7 mM ABTS, 2.5 mM potassium persulfate and 10 mL of distilled water were mixed and incubated in the dark at room temperature for 16 h before use. This solution was diluted with MeOH to an absorbance of 0.7 ± 0.02 at 734 nm. After the addition of 20 µL of extract or Trolox standard to 200 µL of diluted ABTS solution, the absorbances were recorded at 6 min after mixing. Methanolic solutions of known Trolox concentrations were used for calibration. The results are expressed as mg Trolox equivalents (eq)/g sample.

The DPPH antioxidant activity assay was performed by the Brand-Williams et al. (1995) [59] method with slight modifications. A MeOH solution containing 500 µmol of DPPH was prepared. After adjusting the blank with MeOH, an aliquot of 20 µL of extract was added to 200 µL of this solution. After 30 min in the dark, the absorbance at 515 nm was read with the spectrophotometer. The results are expressed as mg Trolox eq/g sample.

#### 2.2.6. HPLC Analysis

Sample analysis was performed on a liquid chromatograph Alliance 2695, equipped with a 2998 Diode Array Detector (Waters, Milford, MA, USA) and Software Empower 3. The separation was carried out on a 5 micron (100 Å, 250 × 4.6 mm) C-18 reverse phase Kromasil column (Ale, Bohus, Sweden) at room temperature. The mobile phase was phosphoric acid at 5 mM (solvent A) and MeOH (solvent B), at a flow rate of 1 mL/min. The elution gradient was as follows: a linear gradient of 85–80% solvent B (0–5 min), 60% B (6–10 min), 70% B (11–20 min), 80% B (21–25 min) and, finally, 85% B (26–30 min). The injection volume was 20 µL, and the 5CQA was detected at a wavelength of 325 nm. This method was adapted from Fujioka and Shibamoto (2008) [60]. Sample chromatograms were compared with those of the 5CQA standard for identification. The measurements were carried out in triplicate. Instrumental calibration: Eight different levels of concentration were employed for 5CQA. The Pearson correlation coefficient (r) was calculated to estimate the type of adjustment of the experimental points in the calibration curve, and subsequently, statistical analyses with Student’s t-test [61] and variance analysis were performed, to verify its significance.

### 2.3. CS Toxicological Analysis

Aflatoxins B1, B2, G1 and G2 were quantified by the method of QuEChERS extraction and ultra-high liquid chromatography tandem mass spectrometry (UPLC-MS/MS) detection [62].

An acute oral toxicity test was performed following the procedure described in the OECD 425 guidelines [63]. Briefly, five female mice, Balb-c strain, 9 weeks old, were used. They were administered 2000 mg/kg of body weight of the aqueous extract of CS, in a single dose, with a cannula, with a 4 h food fast but not water fast. Under the conditions of a temperature of 24 ± 1 °C and photoperiod of 12 h light/12 h darkness, mortality and toxicity signs were registered daily, and weight was measured weekly. Animal experimentation was carried out in accordance with the Official Mexican Method NOM-062-ZOO-1999 [64]; in addition, the protocol was authorized and reviewed by the Internal Committee for the Care and Use of Laboratory Animals of CIATEJ (code 2019-002A).

### 2.4. Product Development

#### 2.4.1. Extrusion Cooking

Ingredient mixes of cornmeal (CM), amaranth flour (AF) and CS were weighed and then mixed in a Kitchen Aid mixer (St. Joseph, Michigan, MI, USA). The mixtures were conditioned to adjust them to 21.0 ± 1.0% of moisture content, placed into plastic bags and maintained under refrigeration for 48 h before processing. Sixteen samples in total were prepared. In each treatment, 300 g of sample was used. 

Extrusion trials were performed using a Brabender single screw extruder (Plasti-Corder 815808, Duisburg, Germany). The barrel diameter and D/L ratio were 475 mm and 19/25, respectively. A screw configuration with a 3:1 compression ratio was used. The exit diameter of the circular die was 2 mm. A vertical dosing screw feeder (628456, Duisburg, Germany) was used for feeding the conditioned mixtures. The process conditions were set as follows: a feed rate of 40 g/min, a screw speed of 80 rpm, and three barrel temperatures—120 °C at the feed entry, 130 °C at the middle and 140 °C at the die exit. The pressure, material temperature and torque were monitored during the extrusion runs. The extrusion conditions were obtained by preliminary tests (data not shown).

Extrudates were left to cool at room temperature for about 30 min. Moisture content was determined [44]. The extruded products were subsequently baked at 60 °C for 2 h, until a moisture content of 5.4 ± 0.3% was achieved, packaged in plastic bags and stored at room temperature until analysis.

#### 2.4.2. Water Solubility Index (WSI)

The method of Anderson et al. (1970) [65] was used. In brief, 2.5 g of sample was added to 30 mL of distilled water at 30 °C, in centrifuge tubes, and shaken on a rotary shaker (Roto-Shake Genie, Bohemia, New York, NY, USA) for 30 min. They were then placed in a centrifuge (Universal 320 R Hettich, Tuttlingen, Germany) run at 4000 rpm for 10 min. The supernatant liquor from each tube was transferred into aluminum trays to be oven dried at 80 °C for 24 h. As the WSI, the amount of dried solids recovered by evaporating the supernatant from the water absorption test just described is expressed as the percentage of dry solids. Analyses were carried out in triplicate.

#### 2.4.3. Experimental Design

An extreme vertices mixture design [66] was used, varying the amounts of AF (50–98%), CM (0–45%) and CS (2–15%). The proposed range of CS was determined according to previous studies in which agri-food industry by-products were incorporated into extruded foods [30,31,32,33]. Figure 2 shows the experimental region of the extreme vertices mixture design, where the 16 points of the experimental runs are indicated.

### 2.5. Statistical Analysis

The STATGRAPHICS Centurion XV package (Statpoint Technologies; Warrengton, VA, USA) was used for the data analysis of the analytical method, as well as for design and analysis of the extreme vertices mixture design. Statistically significant differences between values were determined at the *p* < 0.05 level [66]. The results are expressed as the mean values ± standard errors of the three separate determinations.

## 3. Results and Discussion

### 3.1. CS Characterization and Microbiological Quality

#### 3.1.1. Bromatological Composition

The results of the bromatological analyses for CS are shown in Table 1. The protein content in CS (15.09% w/w) was minor compared to previous values reported, 18.6% w/w [22], and 18.69% w/w [9]. The low fat content in CS (1.99% w/w) was similar to a value cited before, 2.2% w/w [22], and lower than those presented by other authors, 3.78% w/w [9]. The ash, carbohydrate and moisture values obtained were similar to those published before [6,9,22]. The dietary fiber content was 67.6% w/w, which is superior to that published, 54.11% w/w [9] and 62.4% w/w [22]. The differences in the contents of the nutrients could be due to the origins of the coffee beans. 

#### 3.1.2. Microbiological Determinations

There is no regulation for the microbiological parameters for CS, as it is a coffee industry by-product; however, in the Mexican Regulation for roasted coffee [67], less than 3 CFU/g of *E. coli* is specified; thus, by comparison, the CS result shows an acceptable level, and when comparing the results obtained for the remaining microbiological determinations with the Official Mexican Regulation for cereals and their products [68]—because the chemical composition of CS is similar to that of cereals—the results obtained are within the parameters, as shown in Table 2.

#### 3.1.3. Antioxidant Capacity and Total Polyphenol Content

The DPPH assay is based on the change of the blue-violet color towards pale yellow due to the reaction with antioxidant substances. The antioxidant capacity of the sample according to the DPPH method was 33.23 ± 0.02 µM Trolox/g of CS (dry basis, db). In another study, 21.35 ± 0.39 µM eq Trolox/g (db) was reported [9]. The antioxidant capacity according to the ABTS method was 3.45 ± 0.02 mM Trolox/100 g of CS (db). Some authors have published values for CS of 1.92 mM Trolox/100 g [22], and 2.12 ± 0.4 mM Trolox/100 g dry matter [15], which are consistent with results found in this work. 

The total polyphenol assay provides an approximation of the total amount of polyphenols in the sample. In present work was obtained 16.48 ± 6.6 mg GAE/g of CS (db). This was similar to what has been already reported, 16.1 ± 1.2 mg/g of CS [11]. These results suggest the possibility of recycling CS in a new food product as a contribution of bioactive compounds.

#### 3.1.4. Quantification of 5CQA

##### Method Performance

According to Regulation (EC) No. 333/2007 [69], if an analytical method includes an extraction step, the result of the analysis must be corrected based on the recovery, so the level of recovery was calculated. The efficiency of the extraction of 5CQA was 87.01% of recovery. The determination coefficient (r^2^) was 0.99, which demonstrates the linearity of the calibration curve for the 5CQA at eight concentration levels in the range of 10–500 µg/mL. The instrumental detection (LOD) and quantification (LOQ) limits for 5CQA were determined based on the signal-to-noise ratios of 3 and 10, respectively, using the weighted parameters [61], thus obtaining an LOD of 3.311 µg/mL and LOQ of 11.037 µg/mL.

##### 5CQA Content in CS

The concentration of 5CQA extracted from CS was 499.03 ± 7.45 mg of 5CQA/kg of CS (db). An amount of 198.9 ± 6.6 mg of chlorogenic acid/100 g of CS was reported [11], which is four times higher than the concentration obtained in this work. Meanwhile, others studies have shown contents of 1.0 ± 0.0 to 1.7 ± 0.1 mg of chlorogenic acid/g of CS [18], 9.4 ± 2.6 mg of 5CQA/g extract of CS [19] and 89.83 ± 0.64 mg of 5CQA/g of dry extract of CS [20]. The difference in the contents of 5CQA in the CS could be due to the nature of the coffee beans, their origins, the extraction methods, and the processes of coffee roasting. During this process, when the temperature is higher than 160 °C, a series of exothermic and endothermic reactions take place; the bean become light brown, its volume increases considerably and the detachment of CS occurs. The chemical reactions responsible for the aroma and flavor of roasted coffee are triggered at approximately 190 °C. These reactions are interrupted at the desired point based on the bean color or programmed time [70,71,72]. At temperatures between 150 °C and 170 °C the decrease in 5CQA content starts to speed up [72]. Therefore, as the beans (and the CS) stay longer in the roaster, where high temperatures are present, the content of 5CQA considerably diminishes. This could explain the concentration of 5CQA obtained in the CS.

### 3.2. Toxicological Aspects

The negative impact on human health of aflatoxins, especially because of their carcinogenicity, shows the importance of carrying out their quantification [73]. The aflatoxin quantification yielded the following results: aflatoxin B1 < 0.20 ppb, B2 < 0.06 ppb, G1 < 0.20 ppb and G2 < 0.06 ppb. The maximum admissible levels in food, in the European Union, for the sum of the four aflatoxins (B1, B2, G1 and G2) have been set from 4 µg/mL to 15 µg/mL, depending on the type of food (peanuts, nuts, dried fruits and their by-products, and cereals and their by-products) [73]; thus, the sum of the four aflatoxins for CS was below these limits.

For the acute oral toxicity test [63], a single dose administration of aqueous extract (CS) at 2000 mg/kg, was provided by esopharingeal cannulation. Normal behavior was recorded daily in the mice, with normal postural reflex and hygiene habits as well as food and water consumption as appropriate for the species. There were no clinical abnormalities. During the test period (14 days), no signs of evident toxicity or mortality of the experimental mice were observed. The results obtained allow us to affirm that the LD50 is above 2000 mg of CS/kg body weight.

The characterization of the CS and its toxicological evaluation allowed the evaluation of its potential as an ingredient for the food industry, confirming that it is a source of bioactive compounds (including 5CQA), dietary fiber and protein, and low in fat, and that its consumption is safe, so it can be considered for food development.

### 3.3. Product Development

The CS was totally incorporated into a food product to recycle this by-product without generating a new by-product derived from the subsequent process; therefore, an extruded ready-to-eat cereal-based food was developed. 

#### 3.3.1. Product Formulation

The product formulation was developed using the parameters obtained from the bromatological analyses of the three raw materials. The composition obtained for corn was 7.57% protein, 1.24% ashes, 2.22% fats, 77.46% carbohydrates and 7.49% corrected dietary fiber [74], and that for popped amaranth was 15.60% protein, 2.88% ashes, 7.97% fats, 73.55% carbohydrates and 9.41% dietary fiber; the CS composition is described in Section 3.1.1.

An extreme vertices mixture design was used to determine the best combination of the three raw materials—CS, CM and AF—that minimized the WSI and maximized the 5CQA content. The proposed formulations were designed to be classified as foods with high fiber contents and sources of protein, in accordance with the established Regulation (EC) No. 1924/2006 [75], where a food with a high fiber content is one that has a minimum of 6 g of fiber/100 g of product, and a food source of protein is one in which protein contributes at least 12% of the total energy value, which was verified by performing a theoretical calculation using the values obtained from the bromatological analysis of the raw materials, according to the formulations obtained through the mixture design. Table 3 shows the formulations proposed by the mixture design and the results for the dietary fiber and corresponding percentage of energy contributed by proteins, satisfying both requirements.

#### 3.3.2. WSI and 5CQA Content in Extruded Products

Extrusion cooking was accomplished. The WSI and 5CQA content were determined for the extrudates; the results are exhibited in Table 3.

The WSI is related to the quantity of soluble molecules, which is related to dextrinization. Thus, the WSI can be used as an indicator for the degradation of molecular compounds and measures the degree of starch conversion during extrusion [29]. The WSI of the extrudates was influenced by the quadratic effect of the raw materials. The adjusted R-square value was 0.90. The CM effect was more important for the decrease in this property. In Figure 3, a decrease in the WSI with an increase in the CM content can be observed. The reduction in starch degradation lowers the WSI, which increases the bowl life of breakfast cereals and reduces the undesirable powdery mouthfeel of extruded snacks [28].

The 5CQA content of extrudates was influenced by the quadratic effect of the raw materials. The adjusted R-square value was 0.97. The CS effect was more important for the increase in this property. In Figure 4, an increase in the 5CQA content with an increase in the CS content can be observed, as expected since this ingredient is the source of this bioactive compound, as shown in Section 3.1.4.

The optimization of the formulation with the response variables WSI and 5CQA content, using the desirability function [66] in the indicated region, maximizing the 5CQA content and minimizing the WSI response, showed that the optimal values for the studied components were 35% CM, 50% AF and 15% CS. The overall desirability was 0.937. The desirability function predicts the response values of the WSI at 24.67 and 5CQA content at 63.41 mg of 5CQA/kg of extruded product, which are similar to the values determined in Table 3 for this formulation. Figure 5 shows the optimized extruded product.

Further research is suggested in order to study the texture properties and sensory evaluation of the extrudates; also, consumer acceptability needs to be explored.

## 4. Conclusions

The CS can be considered as a new food ingredient, which can increase the protein and dietary fiber content of food, in addition to providing bioactive compounds such as 5CQA, which has been shown to exert benefits in the human organism. Up to 15% of CS was incorporated into the extruded product.

To ensure the safety of CS as a food ingredient, the application of good manufacturing and storage practices are recommended, from its collection in the coffee industry. A grinding process is suggested for easy handling.

Although this study contributes to establishing routes for the valorization of CS, it is important to note that studies are still needed to demonstrate the sustainability of this food industry by-product. To incorporate CS in the food production chain, it is suggested to carry out shelf-life tests, studies on logistics issues and/or business opportunity studies for coffee roasters. An assessment of consumer acceptability is also necessary. Furthermore, economic feasibility studies are required.

## Figures and Tables

**Figure 1 foods-09-01008-f001:**
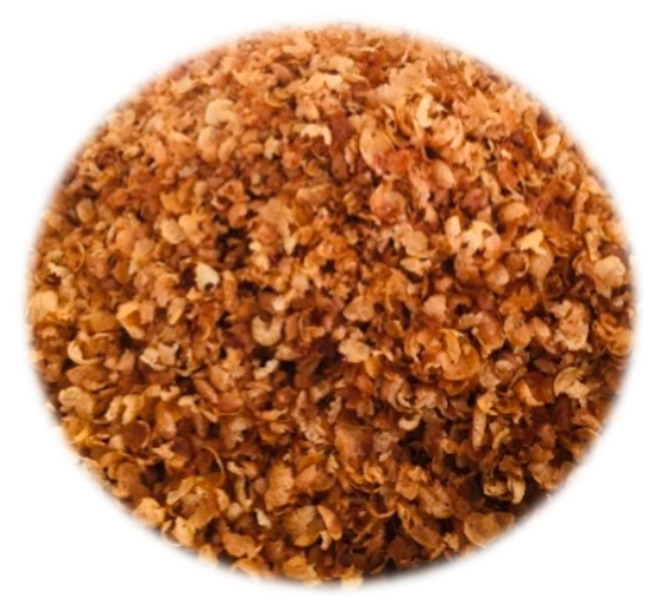
Coffee silverskin from coffee roasting process.

**Figure 2 foods-09-01008-f002:**
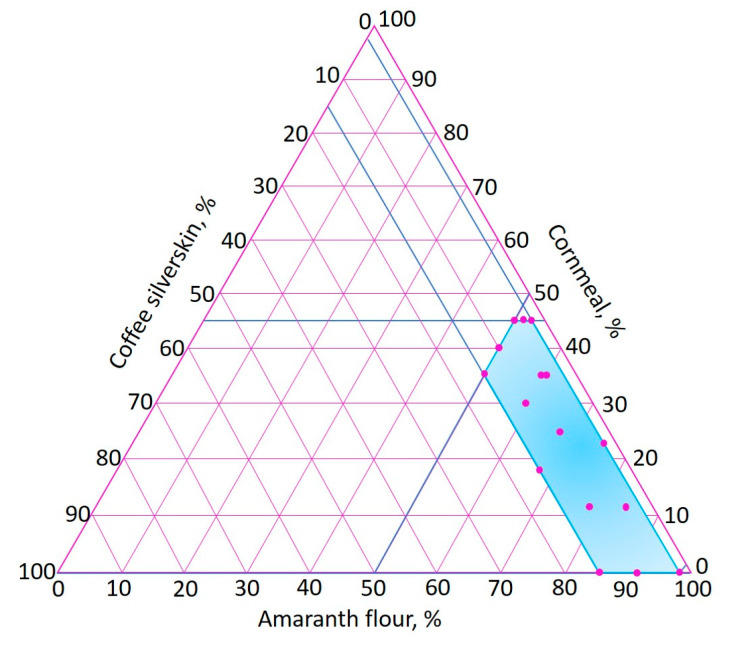
Experimental region of the extreme vertices mixture design.

**Figure 3 foods-09-01008-f003:**
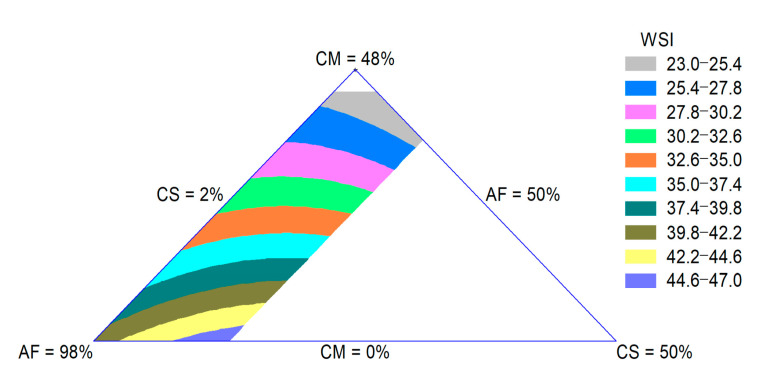
Contour plot for WSI of extruded product formulated with corn meal (CM), amaranth flour (AF) and coffee silverskin (CS).

**Figure 4 foods-09-01008-f004:**
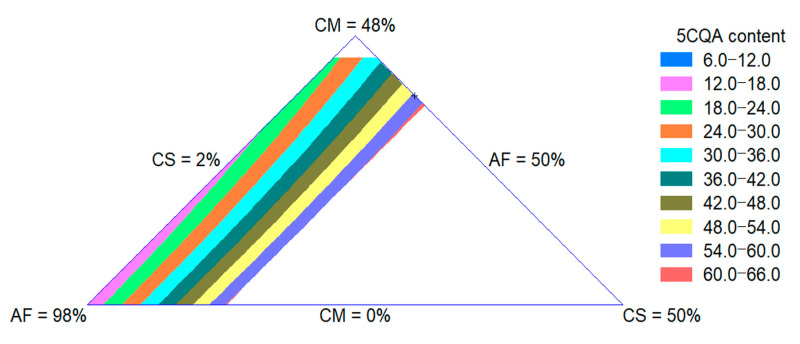
Contour plot for 5CQA content of extruded product formulated with corn meal (CM), amaranth flour (AF) and coffee silverskin (CS).

**Figure 5 foods-09-01008-f005:**
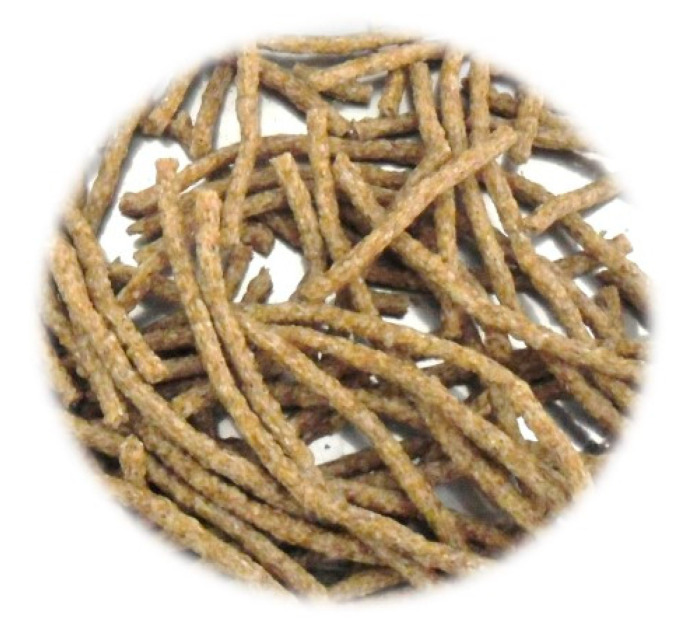
Optimized extruded product.

**Table 1 foods-09-01008-t001:** Chemical composition of coffee silverskin.

Parameters	CS %
Moisture	7.2
Protein	15.09
Ashes	5.55
Fats	1.99
Carbohydrates	70.17
Dietary Fiber	67.6

**Table 2 foods-09-01008-t002:** Microbiological results (CFU/g).

Parameters	CS	Roasted Coffee ^1^	Wholemeal Flour ^2^
Aerobic mesophilic bacteria	1400	ns	500,000
Total coliforms	40	ns	500
Molds	45	ns	500
Yeasts	<10	ns	ns
*Salmonella spp*	Absence	ns	ns
*Escherichia coli*	<3	<3	ns
*Staphylococcus aureus*	<100	ns	ns

^1^ Maximum level suggested for roasted coffee for Mexico by the Secretaría de Economía. ^2^ Maximum level for cereal additives for Mexico by the Secretaría de Salud. ns—not specified.

**Table 3 foods-09-01008-t003:** Extreme vertices mixture design. Theoretical values for dietary fiber and protein energy contribution. WSI and 5CQA concentration determined in extruded product.

CS:CM:AF ^1^ %	Dietary Fiber ^2^ g/100 g Product	Protein Contribution to the TEV ^2,3^ %	WSI ^4,6^	5CQA ^5,6^ mg/kg
2:0:98	10.3	14.6	39.85 ± 0.86	13.64 ± 0.24
15:35:50	20.7	12.9	27.01 ± 0.11	63.60 ± 1.48
15:0:85	16.1	14.9	46.34 ± 0.88	62.94 ± 1.12
5:45:50	17.5	12.1	21.73 ± 0.79	28.77 ± 0.73
2:45:53	16.2	12.1	26.59 ± 0.87	27.01 ± 0.75
4.9:12.5:82.6	13.2	14.0	37.76 ± 0.16	25.50 ± 0.68
11.4:30:58.6	18.4	13.1	29.68 ± 0.22	47.35 ± 1.08
11.4:12.5:76.1	16.1	14.1	41.45 ± 0.09	51.58 ± 1.49
6.4:35:58.6	16.8	12.8	29.91 ± 1.42	33.09 ± 0.61
4.9:35:60.1	16.2	12.7	23.59 ± 0.62	29.58 ± 0.84
8.5:0:91.5	13.2	14.7	44.79 ± 0.52	34.11 ± 0.90
2:22.5:75.5	13.2	13.4	33.23 ± 0.71	13.57 ± 0.12
15:17.5:67.5	18.4	13.9	31.86 ± 0.30	59.82 ± 0.51
10:40:50	19.1	12.5	22.54 ± 0.61	49.39 ± 0.78
3.5:45:51.5	16.8	12.1	23.52 ± 0.84	23.18 ± 0.39
7.8:25:67.2	16.2	13.4	31.55 ± 0.37	34.70 ± 0.18

^1^ CS:CM:AF, coffee silverskin/cornmeal/amaranth flour; ^2^ calculated values for each mixture; ^3^ TEV, Total Energy Value; ^4^ WSI, Water Solubility Index; ^5^ 5CQA, 5-caffeoylquinic acid; mg/kg, mg of 5CQA/kg extruded product; ^6^ Measured values.

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
