# Peer review of "Evaluation of the Use of a Coffee Industry By-Product in a Cereal-Based Extruded Food Product"

_foods, 2020, doi:10.3390/foods9081008_

Round 1
Reviewer 1 Report
As far as the results of your experiments and the DOE are concerned, I have not many requests.
My principal request is connected to an evaluation of the scalability of the technology you have proposed in this article. In fact, many researches have been done at a laboratory scale (as yours) as regard to the valorization of food waste. I would like you to insert a specific paragraph about the feasibility study of an application at a pilot plant or at industrial level, considering in particular the scalability of your technology.
As you can find in an interesting new article, which you should mention in the introduction section (https://doi.org/10.1016/j.biortech.2020.123575), the technical feasibility and the profitability of each technology will define its future real application at an industrial level. It is nowadays extremely important that your research will allow future real applications at an industrial level.
Based on this request, please revise also the abstract, the introduction and the conclusion sections adding some considerations about it.
Author Response
Please see the responses attached.

Reviewer 2 Report
The manuscript evaluate the characteristics of coffee silver skin and its addition to extruded cereal base snacks as a source of fibres and proteins. Overall, the manuscript is interesting and the multiple analysis performed, including toxicity in mice, showed that this coffee bio products is safe for consumption.
The points I would like to address are:
- Many parts in the text required a better structure, as the sentences are very long and with many commas. For example, lines 48-54; 70-81. The clarity of the text can be improved by making the sentences shorter.
- Line 137: specify the frequency used for sonication
- In many cases, the references are cited both as an author name and as a number. Please rephrase the construction in order to use the number only. For example line 210-202; 242-245 etc
- I think the structure of the paper could benefit of some adjustment in the structure of material and methods as well as the results. The three main area investigated are -proximate composition and microbiological quality - Toxicological aspect -Product development.
Could be helpful to separate both, the materials and the results section, into subgroups of these three major areas.
- Section 3.2 doesn’t really contribute to the discussion. Is more an extension of the introduction.
- It will be interesting perhaps to have some picture of the developed extruded products. If available.
- Conclusion needs to be expanded. Especially in the case of using CS for product development it will be advisable to suggest further studies in order to check for consumer acceptability and sensory properties of the food.
Author Response
Please see the response attached.

Round 2
Reviewer 1 Report
Authors answered all my questions.
The article is improved also thanks to the new structure.
It could be accepted in this form